# Early Childhood Caries—Prevalence, Associated Factors, and Severity: A Hospital-Based Study in Riyadh, Saudi Arabia

**DOI:** 10.3390/healthcare12141376

**Published:** 2024-07-10

**Authors:** Ashokkumar Thirunavukkarasu, Sultan Fadel Alaqidi

**Affiliations:** 1Department of Family and Community Medicine, College of Medicine, Jouf University, Sakaka 72388, Saudi Arabia; salaqide@moh.gov.sa; 2Department of General Dental Services, Dental Clinic Complex, North Riyadh 123216, Saudi Arabia

**Keywords:** early childhood caries, prevalence, associated factors, severity

## Abstract

Early childhood caries (ECC) is not merely a problem of the tooth; rather, it has negative impacts on the child’s growth and development and oral health-related quality of life. The prevalence of ECC varies widely, and more variance is observed between countries than continents, and it changes over time. The current cross-sectional study aimed to determine ECC’s prevalence, associated factors, and severity. This study was conducted in the Dental Clinic Complex, North Riyadh, KSA. The study sample included 306 participants between 36 and 71 months of age. A questionnaire was developed to collect information from participants. The decaying and filled teeth (DMFT) index was used to estimate the prevalence of caries. The study results revealed the prevalence of ECC to be 76% among 36- to 71-month-old children in North Riyadh. ECC is influenced by socioeconomic factors like maternal occupation, education, oral hygiene habits, and dietary patterns. Also, it was found that exclusively breastfed children have lower ECC odds (68.1% vs. 83.6%, *p* = 0.001), while the intake of more than three between-meal sugar-containing snacks/beverages per day increases the likelihood of ECC (63.5% vs. 79.8%, *p* = 0.006). The researchers recommend encouraging breastfeeding as it correlates with lower ECC prevalence, in addition to establishing effective oral hygiene practices for preschool children.

## 1. Introduction

According to the Global Burden of Disease (GBD) 2017 report, dental caries in permanent teeth has the highest prevalence out of all diseases [1]. A systematic review conducted of reports from 67 countries suggests that in most developed countries, the prevalence varies between 1 and 12%, whereas in underdeveloped countries and among underprivileged people, the prevalence is considered as high as 70%. Sweden and Italy exhibit a significantly low prevalence rate of ECC, with rates of 11.4% and 7–19%, respectively [2]. In contrast, the prevalence is higher several Middle Eastern countries, such as Palestine, which recorded a higher incidence rate of 76%, and the UAE, which recorded 83% [1].

National surveys of different countries, including China, Saudi Arabia (KSA), Brazil, and India, show inconsistent prevalence rates [3,4,5,6]. ECC is associated with a combination of factors, including enamel growth abnormalities and the presence of acid-producing bacteria in a complex biofilm, which occurs when sugar is consumed often by children. Dental caries in preschoolers is a major problem that has huge societal expenses and has an influence on children’s quality of life [5,7]. Evidence from around the world shows that ECC is still rather common, but it is rarely addressed [1]. The severity of ECC is associated with socioeconomic variables, dietary habits, tooth-brushing habits, attending rural private schools, two main meals/day, and unsupervised tooth brushing [8,9,10]. The impact of dietary practices, including the frequency and type of sugar intake, has been increasingly recognized as a crucial determinant of ECC’s prevalence [11]. There is a growing body of evidence suggesting that ECC not only affects oral health but also has broader implications for general health, cognitive development, and quality of life among children [5,12,13,14]. The general decayed, missing, and filled teeth (DMFT) index shows a medium severity level according to the World Health Organization classification, where girls present a higher level of risk than boys [15,16].

ECC affects children’s social development, such as poor school attendance and poor performance at school [17]. According to the American Academy of Pediatric Dentistry [18], only 45% of countries of the United Nations have data on ECC, while the KSA has limited nationwide survey data except for a few uncoordinated individual studies [4,15,19]. A cross-sectional study was conducted by Acuña J.E.C. et al. in Quito, Ecuador, among 557 children attending childcare centers, which found that the prevalence of early childhood caries was 59.61%. Children affected by ECC presented a mean age of 2.83 years, and the group most affected was those of 2 years old (35.94%) [20]. Boys have suffered more than girls (53.92% vs. 46.08%). Almarshad L.K. et al. conducted a cross-sectional study among 383 preschoolers aged 36–71 months in the KSA and found that the ECC prevalence was 72.6%, with a mean DMFT score of 4.13 (±3.99) and a mean decayed, missing, and filled surfaces (DMSF) score of 7.0 (±9.1) [15]. 

Lara J.S. et al. carried out a cross-sectional study to determine the impact and severity of ECC in 409 Mexican preschool students and found significant linear trends among the caries level and severity, socioeconomic variables, dietary habits, and tooth-brushing habits [21]. Attending rural private schools, two main meals/day, and unsupervised tooth brushing were found to increase the risk of severe ECC [22]. 

Adequate information regarding the key characteristics of ECC, the occurrence, and the risk factors in Riyadh, the KSA, is necessary because this condition poses a major threat to children’s well-being and health. Recent studies have highlighted the significant role of early preventive dental care and oral health education in reducing the prevalence of ECC among preschool children [23,24]. Thus, knowing localized determinants of caries’ prevalence may help improve the accessibility of feasible preventive steps. Furthermore, this study can provide relevant information concerning the scarcity of available data on pediatric oral health within the region, which can be used for future research and healthcare planning. Therefore, this research was conducted to identify the prevalence, associated factors, and severity of ECC.

## 2. Methods and Materials

### 2.1. Study Description

This cross-sectional study was carried out in the Dental Clinic Complex, North Riyadh, the KSA. This study was conducted from March 2024 to May 2024. Participants were those attending the outpatient and inpatient departments of Dental Clinic Complex, North Riyadh, the KSA, and they were aged between 36 and 71 months. The exclusion criteria included children with known mental illness, children with known dental anomalies, children using continuous medication, and children suffering from chronic disease. 

### 2.2. Sampling Description

The calculated sample size was 306, with a 95% confidence interval and 5% margin of error, using the following formula: sample size (n) = z^2^pq/d^2^ = 306 where z = 1.96 in 95% CI. *p* = 0.726, Prevalence is 72.6% [15] q = 1 − *p* = 1 − 0.726 = 0.274 d = 0.05 as margin of error 5%. We applied a convenience sampling method to collect data from the participants. Convenience sampling was chosen due to its feasibility and efficiency in a healthcare setting. This method allowed us to quickly and effectively recruit participants who were already present in the dental clinic, thereby maximizing the response rate and ensuring a sufficient sample size within the limited study period. With this method, the parents or legal guardians of the children were requested to participate during their visit to the dental complex.

### 2.3. Clinical Examination

Taking five pictures of each child’s teeth using a smartphone camera was the standard practice after the pediatric dental exams. Four dental reviewers—not the same people who looked at the children’s teeth—analyzed the intraoral photos to check for dental caries. All the examiners had undergone a standardized training program to ensure uniformity in caries detection. In cases where there was a disagreement on a diagnosis, we had a group discussion with the standard criteria to reach an agreement. If a consensus could not be reached, the case was reviewed by a senior pediatric dentist, whose decision was considered final by the research team. We assessed the photographic method’s sensitivity, specificity, and inter-rater reliability agreement to evaluate its diagnostic performance compared to the benchmark visual dental examinations. The decayed, missing, filled teeth (DMFT) index was used to estimate the prevalence and severity of ECC in the study participants. The DMFT index is specifically designed for primary teeth and is appropriate for assessing dental caries in children aged 36–71 months. We classified the severity of ECC based on the DMFT index as follows: mild ECC (type I)—DMFT score of 1–3, moderate ECC (type II)—DMFT score of 4–6, and severe ECC (type III)—DMFT score of 7 or higher.

### 2.4. Data Collection Tool

A questionnaire was developed by the research team and dental public health experts to collect information from participants. The questionnaire gathered sociodemographic information (Section 1), information regarding associated factors (Section 2), and information on the severity of early childhood caries (Section 3). Section 1 inquired the participants’ age, gender, residence, occupation (father and mother) and education (father and mother). Section 2 asked about the child’s dental care activities. The questionnaire was pre-tested two different times on 20 parents who were not included in the main study. Then, the parents (father or mother) of all the recruited children were interviewed using this questionnaire after obtaining informed consent. The Cronbach’s alpha value of the developed questionnaire was 0.82. Participation was voluntary, and responses remained anonymous (serial numbers replaced names). Then, the recruited children who fulfilled the inclusion criteria underwent intraoral examination.

### 2.5. Data Management and Statistical Analysis

The information was recorded in an interviewer-administered questionnaire. All the data were entered, checked, rechecked, and scrutinized by the principal investigator following standard procedures and were analyzed using SPSS 25.0 (Statistical Program for Social Science) software. The results of the study and statistical analysis were presented in the form of text, tables, bar diagrams, graphs, and charts. Categorical variables were displayed as the frequency and percentage. Numerical (continuous) variables were summarized as the mean ± SD. Group comparisons were carried out using appropriate statistical analyses (Chi-square test). The researcher collected the data via the questionnaire, which was then checked and analyzed using the SPSS program. The results were clarified, and the clinical examination was related to the results of the questionnaire. 

## 3. Ethical Consideration

In accordance with the Declaration of Helsinki Principles for Medical Research Involving Human Subjects (1964), the participants of the study were provided with oral information regarding the study’s design, objectives, and their entitlement to withdraw from the project at any point and for any reason. The study comprised parent/legal guardian participants who provided informed and voluntary consent. We obtained ethical approval from the Jouf University Bioethics Committee (approval no.: 2-08-45).

## 4. Observations and Results

This cross-sectional study was conducted in the Dental Clinic Complex, North Riyadh, the KSA, from March 2024 to April 2024. The objective of the study was to determine the prevalence, associated factors, and severity of early childhood caries among 36- to 71-month-old children. A total of 306 participants were enrolled in the study by convenience sampling method from those attending outpatient and inpatient departments, and were aged between 36 and 71 months, after considering the inclusion and exclusion criteria.

Table 1 illustrates the study participants’ background characteristics, ECC status, and associated factors. The mean age of the study participants was 58.86 months. Most participants were 60–71 months of age (51%), followed by 48–59 months (40.2%). However, only 8.8% of the participants were from the age group 36–47 months. Also, the table indicates that the majority of participants were female (56.2%). Additionally, a significant proportion of participants resided in urban areas (60.1%). The majority of families reported monthly expenditures in the range of USD 2000–3000 (46.1%), followed by USD 1001–2000 (37.3%). Moreover, the table illustrates that the majority of mothers were unemployed or working in the private sector. Furthermore, the prevalence of ECC across different demographic variables was 76.4%. Notably, ECC was most prevalent in the age group of 60–71 months (82.7%), followed by 48–59 months (78.9%), with a statistically significant *p*-value of <0.01. Gender-wise, the ECC rates were almost equal between males and females (75.4% vs. 77.3%). Rural participants exhibited a higher prevalence of ECC (84.4%) compared to their urban counterparts (71.2%) (*p* = 0.007). Additionally, those with a monthly expenditure of less than USD 1000 had the highest ECC prevalence at 94.1%, with a statistically significant *p*-value of 0.006. Furthermore, the mother’s occupation and education were also significantly associated with ECC (*p* = 0.001).

The present study found that the prevalence of ECC among the participants was 76.4%. Figure 1 shows that the prevalence of ECC was higher in females within the age group 36–47 months, reaching 33.3%. In contrast, the age groups of 48–59 months and 60–71 months were approximately equal in both genders (79% vs. 78.7% and 81.7% vs. 83.3%), respectively.

Figure 2 illustrates that in the studied population, 115 (37.6%) of ECC cases were classified as Type I, followed by 27 (21.9%) Type II cases, with Type III comprising 52 (17%) cases.

Table 2 delineates the association between infant feeding practices and the prevalence of ECC within the study population. The data reveal that children with a history of exclusive breastfeeding (6 months) exhibited a significantly lower ECC rate of 68.1%, while those with a history of bottle feeding had a prevalence of 83.6% (*p* = 0.001). Furthermore, children ceasing breastfeeding or bottle feeding at more than 18 months of age demonstrated a heightened ECC percentage of 100%, followed by the age group of 12–18 months (77.8%). Notably, those children who habitually walked around drinking from a bottle or cup had a higher prevalence of ECC (82.3%).

Table 3 provides an insightful glimpse into the distribution of the study participants based on various infant teeth care habits and their correlation with the presence of ECC. Among the assessed habits, including thumb or finger sucking and mouth breathing, no significant differences in the ECC prevalence were observed between children with these habits and those without (77.5% vs. 76.0%). However, the presence of a family history of ECC (78.6% vs. 75.4%), residence in a fluorinated community (76.1% vs. 77.5%), and the availability of a healthcare center with dental providers (76.9 vs. 76.2%) in the community had slightly higher prevalence of ECC; however, they did not exhibit statistically significant associations with the ECC prevalence.

When exploring oral hygiene practices, the tooth-brushing frequency, assistance with brushing, and the frequency of flossing showed no significant differences in terms of the ECC rates. The use of fluorinated toothpaste also did not demonstrate a significant association with ECC. Furthermore, the age at which tooth brushing commenced, categorized as <12 months, 12–24 months, and >24 months, did not reveal any statistically significant differences in the ECC prevalence.

## 5. Discussion

The present study found that more than three-fourths (76.4%) of participants had ECC. Some studies conducted in the KSA and other Asian countries also indicated a higher prevalence [15,25,26,27]. In contrast, some studies from Western countries, including the National Health and Nutrition Examination Survey (NHANES) in the USA, demonstrated a lower prevalence [28,29]. This could be due to the sugary food consumption among the children in this region [30,31]. For instance, Memena W.A. revealed that only 0.90% of the participants in their study had free sugar consumption according to the World Health Organization (WHO)’s recommendations [30]. The higher level of ECC in this region indicates that this issue remains a significant public health issue and needs to be addressed immediately with different dental preventive approaches. 

We found that the higher age group (60–71 months) had a higher proportion of ECC (82.7%, *p* = 0.001) than the other age groups. Our findings are similar to some other studies [7,9,32]. The possible reasons for this group’s higher proportion of ECC could be dietary habits, as they may be more independent in choosing snacks and oral hygiene habits. This suggests the need for targeted preventive strategies and early intervention programs to reduce the incidence of ECC as children grow older. The prevalence of dental caries was strongly correlated with the child’s age and the frequency of dental checkups [7,32,33]. It is worth mention that younger children sleep more than the higher age group. Sleep-induced salivary flow reduces salivary capacity, shifting the balance toward demineralization [34,35]. Water consumption in infancy, particularly bottled water, may be beneficial due to its washing effect or fluoride content. However, the ECC prevalence increased significantly with increased sugar consumption between meals, consistent with previous reports of a link between prolonged high sugar consumption and ECC in children [9,19]. Studies showed that the caries prevalence at five years was significantly related to tooth brushing less than twice daily during preschool and difficulties in performing the procedure [36,37]. Therefore, health professionals should pay special attention to this issue and assist parents in improving and optimizing their children’s tooth-brushing behavior during preschool.

Similarly, those who lived on the rural side had a significantly higher proportion of ECC (84.4%, *p* = 0.007) than others. This could be due to limited access to dental care, socioeconomic challenges, and dietary habits. Another critical determinant of ECC was the mother’s education. We found that the highest levels of ECC were observed among the children with a lower level of mother’s education. This finding highlights the importance of parental education, policy development, and community-based initiatives to improve oral health outcomes among preschoolers [38]. Similar to other studies, we observed that there was no association identified between gender and the development of ECC [33,39]. 

Regarding the severity of ECC, we found that type I (mild) ECC was the most prevalent at 37.6%, followed by type II (moderate) at 21.9%, with type III (severe) comprising 17% of cases. Our findings of mild ECC are similar to those of other studies [40,41]. However, we found a notably higher level of severe ECC. The possible reason for this higher proportion of severe ECC prevalence could be due to the study setting. The present study was conducted in a dental clinic. As seen from these assertions, the implications of the findings are the following. Firstly, understanding the distribution of the ECC severity types helps tailor preventive strategies and treatment approaches accordingly. Hence, interventions aimed at early identification and management of mild and moderate ECC might have the potential to slow the deterioration of this disease and the emergence of its complications to enhance children’s oral health [13,42]. Secondly, the higher prevalence of severe ECC cases underlines the need for the implementation of specific measures oriented toward the given community, such as dentist availability, oral health promotion, or epidemiologic preventive measures. Thus, addressing the issues using an appropriate public health approach can lower the burden of ECC and enhance children’s oral health in the community [14,43].

This study explored the relationship between infant feeding practices, oral hygiene habits, and ECC prevalence. Children with a history of breastfeeding exhibited a slightly lower ECC rate, aligning with established recommendations concerning breastfeeding’s protective effect against dental caries [44]. Notably, timely weaning of breastfeeding and cessation of bottle feeding was associated with a lower ECC prevalence, emphasizing the importance of weaning practices in preventing dental caries [45,46]. In contrast, certain oral hygiene practices and habits, such as tooth-brushing frequency, assistance with brushing, and the use of fluorinated toothpaste, did not show significant associations with ECC prevalence [19]. The association between more than three between-meal sugar-containing snacks and the ECC prevalence highlighted the role of dietary habits in oral health outcomes [7,32]. 

## 6. Limitation of the Study

-Cross-sectional design restricts causal relationships.-Self-reported data introduce recall and social desirability bias.-We acknowledge that convenience sampling may introduce selection bias, as the participants may not be representative of the general population. The children visiting the dental clinic are likely to have higher dental health needs, which could potentially overestimate the prevalence of ECC.-Finally, the challenges associated with the detection of proximal caries cannot be ignored.

## 7. Conclusions

This study reveals a 76% prevalence of ECC among 36- to 71-month-old children in North Riyadh, the KSA. The prevalence is higher than in previous research, indicating a significant burden of ECC among preschool children. Type I ECC is prevalent, with a higher prevalence in rural areas. ECC is influenced by socioeconomic factors like maternal occupation, education, infant feeding practices, oral hygiene habits, and dietary patterns. Based on the above-mentioned results, we recommend implementing ongoing research on ECC prevalence to follow the changing trend of prevalence and adjust preventive strategies based on the findings to reduce the high burden, as well as designing educational programs and preventive measures specific to prevalent age and gender groups, in addition to conducting in-depth investigations into the unexpected relationship between family income and ECC prevalence. Furthermore, future studies are to be conducted to evaluate the long-term outcomes of educational programs across different socioeconomic settings and the impact of cultural and behavioral factors. Additionally, investigating the barriers to accessing dental care could provide deeper insights into effective preventive strategies.

## Figures and Tables

**Figure 1 healthcare-12-01376-f001:**
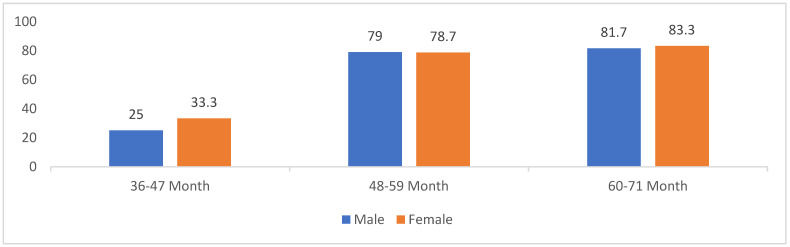
Prevalence among study participants of early onset dental carries by age and gender (n = 306).

**Figure 2 healthcare-12-01376-f002:**
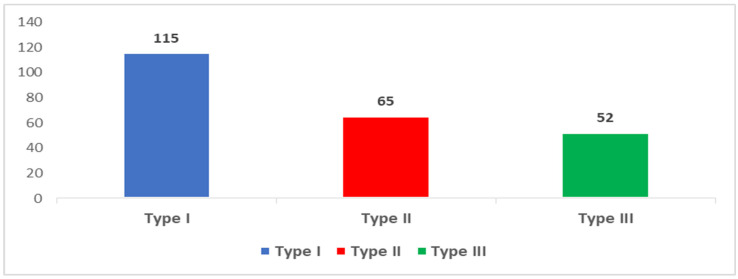
Distribution of study participants by type of ECC (n = 306).

**Table 1 healthcare-12-01376-t001:** The demographic characteristics of the study participants and their early childhood caries (ECC) status (n = 306).

Parameter	Total (306)	Caries-Free (72) n (%)	Early Childhood Caries (ECC) (234) n (%)	*p*-Value
Age Group (Months)				
36–47	27 (8.8)	19 (70.4)	8 (29.6)	0.001
48–59	123 (40.2)	26 (21.1)	97 (78.9)	
60–71	156 (51.0)	27 (17.3)	129 (82.7)	
Mean ± SD (Range)	58.86 ± 7.81 (36–70)			
Gender				
Male	134 (43.8)	33 (24.6)	101 (75.4)	0.689
Female	172 (56.2)	39 (22.7)	133 (77.3)	
Residence				
Urban	184 (60.1)	53 (28.8)	131 (71.2)	0.007
Rural	122 (39.9)	19 (15.6)	103 (84.4)	
Monthly Family Expenditure				
<1000	34 (11.1)	2 (5.9)	32 (94.1)	0.006
1001–2000	114 (37.3)	24 (21.1)	90 (78.9)	
2000–3000	141 (46.1)	44 (31.2)	97 (68.8)	
>3000	17 (5.6)	2 (11.8)	15 (88.2)	
Mother’s Occupation				
Unemployed (Housewife/Student)	104 (34.0)	12 (11.5)	92 (88.5)	0.001
Private Sector/Business	110 (35.9)	20 (18.2)	90 (81.8)	
Service Holder	92 (30.1)	40 (43.5)	52 (56.5)	
Father’s Occupation				
Private Sector/Business	218 (71.2)	51 (23.4)	167 (76.6)	0.930
Service Holder	88 (28.8)	21 (23.9)	67 (76.1)	
Mother’s Education				
Up to Junior High School	154 (50.3)	9 (5.84)	145 (94.1)	0.001
High School	74 (24.2)	52 (70.3)	22 (29.7)	
University	78 (25.5)	11 (14.1)	67 (85.9)	
Father’s Education				
Up to Junior High School	114 (37.2)	27 (23.7)	87 (76.3)	0.001
High School	88 (28.8)	35 (39.8)	53 (60.2)	
University	104 (34.0)	10 (9.6)	94 (90.4)	

**Table 2 healthcare-12-01376-t002:** Distribution of the study participants’ history of infant feeding/dietary variables by ECC status (n = 306).

Characteristic	Carries-Free	ECC	*p*-Value
History of exclusive breastfeeding for 6 months			
Yes	45 (31.9)	96 (68.1)	0.001
No	27 (16.4)	138 (83.6)	
History of bottle feeding			
Yes	66 (23.6)	214 (76.4)	0.588
No	6 (23.1)	20 (76.9)	
Age stopped bottle-/breastfeeding			
<12 months	28 (28.0)	72 (72.0)	0.459
12–18 months	22 (22.2)	77 (77.8)	
>18 months	22 (20.6)	85 (79.4)	
Child walks around drinking from a bottle/cup.			
Yes	20 (17.7)	93 (82.3)	0.071
No	52 (26.9)	141 (73.1)	
Does your child take a bottle to bed?			
Yes	19 (17.6)	89 (82.4)	0.461
No	53 (26.8)	145 (73.2)	
Child has more than 3 between-meal sugar containing snacks/beverages per day			
Yes	23 (36.5)	40 (63.5)	0.006
No	49 (20.2)	194 (79.8)	
Is your child put to bed with a bottle containing natural or added sugar?			
Yes	19 (29.2)	46 (70.8)	0.141
No	53 (22.0)	188 (78.0)	
Does your child continuously take bottle/cup with fluid other than water?			
Yes	17 (27.0)	46 (73.0)	0.28
No	55 (22.6)	188 (78.0)	

**Table 3 healthcare-12-01376-t003:** Distribution of the study participants’ history of infant teeth care habits by ECC status (n = 306).

Characteristic	Carries-Free	ECC	*p*-Value
Does your child have any habits (thumb or finger sucking, mouth breathing)			
Yes	23 (22.5)	79 (77.5)	0.886
No	49 (24.0)	155 (76.0)	
History of ECC in any other family members			
Yes	22 (21.4)	81 (78.6)	0.570
No	50 (24.6)	153 (75.4)	
Child lives in a fluorinated community			
Yes	54 (23.9)	172 (76.1)	0.879
No	18 (22.5)	62 (77.5)	
Community has existing healthcare center with dental provider			
Yes	31 (23.1)	103 (76.9)	0.498
No	41 (23.8)	131 (76.2)	
How many times does your child brush teeth per day?			
One	29 (27.1)	78 (72.9)	0.428
Two	24 (23.8)	77 (76.2)	
Three	19 (19.4)	79 (80.6)	
How many times you assist your child with brushing per day?			
One	25 (23.4)	82 (76.6)	0.959
Two	23 (22.8)	78 (77.2)	
Three	24 (24.5)	74 (75.5)	
How often does your child floss their teeth?			
One	22 (24.2)	69 (75.8)	0.977
Two	21 (22.8)	71 (77.2)	
Three	29 (23.6)	94 (76.4)	
Does your child use fluorinated toothpaste?			
Yes	4 (18.2)	18 (81.8)	0.288
No	10 (28.6)	25 (71.4)	
Age at start of brushing			
<12 months	13 (25.0)	39 (75.0)	0.972
12–24 months	32 (24.6)	98 (75.4)	
>24 months	27 (26.0)	77 (74.0)	

## Data Availability

The raw data supporting the conclusions of this article will be made available by the authors on request.

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
