# Peer review of "Early Childhood Caries—Prevalence, Associated Factors, and Severity: A Hospital-Based Study in Riyadh, Saudi Arabia"

_healthcare, 2024, doi:10.3390/healthcare12141376_

Round 1

Reviewer 1 Report

Comments and Suggestions for Authors

This manuscript is well-written and discusses early childhood caries in Riyadh city. However, I have some suggestions for improvement.

1. The sample size of this study needs to be increased.

2. It would be better to rewrite the introduction section with articles that are more relevant and updated.

3. Please more discuss about your results in discussion part with recent relevant articles.

3. 

Comments on the Quality of English Language

 Minor editing of English language required

Author Response

Dear Reviewer,

Authors’ reply/modifications according to the reviewer 1 comments/suggestions

General:

The authors would like to thank the reviewer for the precious time spent reviewing the paper and his excellent suggestions for improving it. Efforts have been made to modify the paper as per the reviewer’s suggestions and recommendations. The authors will be happy to hear a positive reply. All the points included according to the reviewer’s comments can be seen in track changes.

Specific response to the reviewer’s suggestions:

Kindly find the attached response to each question one by one:

Point 1: This manuscript is well-written and discusses early childhood caries in Riyadh city. However, I have some suggestions for improvement.

Response 1: The authors thank the reviewer for the wonderful positive comments and it encourage the authors in a positive way.

Point 2: The sample size of this study needs to be increased.

Response 2: Thanks for the comment. We understand the reviewer’s comment on the importance of a higher sample size. We calculated the minimum required sample (study participants) using WHO sample size calculator (Cochran’s formula) based on the previous proportion. Hence, we adhered to the minimum sample size obtained for the present study. However, the authors are planning to expand the same study into an exploratory multicentric study with a larger sample size (in accordance with the reviewer’s suggestions).

Point 3: It would be better to rewrite the introduction section with articles that are more relevant and updated.

Response 3: Thank you very much for the valuable comment. We modified the introduction section with the new and updated references according to the reviewer's suggestions.

Point 4: Please more discuss about your results in discussion part with recent relevant articles

Response 4: Thank you very much for the valuable comment. We modified the discussion section with the new and updated references according to the reviewer's suggestions.

The author wishes to thank the reviewer once again for the positive and constructive comments.

Reviewer 2 Report

Comments and Suggestions for Authors

Dear authors

I read your valuable study precisely and I think some clarifications may be helpful:

1. You selected your samples from a dental clinic and probably they were there for their dental problems, most commonly dental caries (due to its prevalence in this age duration). Therefore, your results may overestimate the rate of ECC.

2. What did you do for children with tooth anomaly like amelogenesis imperfecta? did you exclude them from study?

3. What was your plan for situations when the diagnosis of caries was different between your five examiners? 

4. If a participant had lost her or his tooth due to caries, how do you consider it in your study? and when there was a congenital missing?

5. Was there any standard for taking pictures by smartphone, for example resolution, light, distance,...?

6. Are you sure proximal caries are detectable on photographs taken by smart phones?

7. I could not find any description about severity of ECC in your manuscript.

what do you mean of this word?

How did you determine it by questionnaire?

9. What was your sampling technique, convenient (line 86) or simple random (line 129)?

10. I could not find any explanation about type of ECC.

11. I was interested to read about the dft index numbers in different groups in your study.

Comments on the Quality of English Language

It is good.

 In table 3, "exckusive" word should be corrected.

Author Response

Dear Reviewer,

Authors’ reply/modifications according to the reviewer 2 comments/suggestions

General:

The authors would like to thank the reviewer for the precious time spent reviewing the paper and his excellent suggestions for improving it. Efforts have been made to modify the paper as per the reviewer’s suggestions and recommendations. The authors will be happy to hear a positive reply. All the points included according to the reviewer’s comments can be seen in track changes.

Specific response to the reviewer’s suggestions:

Kindly find the attached response to each question one by one:

Point 1: I read your valuable study precisely and I think some clarifications may be helpful:

Response 1: Thank you very much for the precious time spent reviewing our manuscript and valuable suggestions. The authors incorporated/clarified this according to the reviewer’s comments.

Point 2: You selected your samples from a dental clinic and probably they were there for their dental problems, most commonly dental caries (due to its prevalence in this age duration). Therefore, your results may overestimate the rate of ECC.

Response 2: Thank you very much for the comment. The authors acknowledge the reviewer's concern regarding potential sampling bias and overestimate the rate of ECC. The authors would like to clarify that the study was conducted in a dental clinic that also provides preventive dental care services. The implications of this were that not all participants attend the dental clinics with existing dental conditions but also for periodic screening/ preventive care. We agree that there may be some degree of overestimation. Our sample also includes children without active dental issues, offering a more balanced view. Importantly, our study aimed to understand the local determinants of ECC to plan for prevention strategies. The authors welcome any further feedback/comments from the reviewer.

Point 3: What did you do for children with tooth anomaly like amelogenesis imperfecta? did you exclude them from study?

Response 3: Thanks for the comment and for notifying us of the missing inclusion and exclusion criteria details. The authors excluded all children with known dental anomalies. We applied these exclusion criteria to ensure that our sample population represented typical cases of ECC without confounding factors.

Point 4: What was your plan for situations when the diagnosis of caries was different between your five examiners?

Response 4: Thanks for the comment. All the examiners had undergone a standardized training program to ensure uniformity in caries detection. In cases where there was a disagreement on a diagnosis, we had a group discussion with the standard criteria to reach an agreement. If a consensus could not be reached, the case was reviewed by a senior pediatric dentist, whose decision was considered final by the research team. The above-mentioned statement is included in the revised manuscript according to the reviewer’s suggestions and better clarifications.

Point 5: If a participant had lost her or his tooth due to caries, how do you consider it in your study? and when there was a congenital missing?

Response 5: Thank you very much for the comment. We would like to clarify that missing teeth due to caries was considered a caries experience and documented in the DMFT index. In the event of loss of teeth due to congenital anomalies, those children were excluded in the beginning stage itself as a part of the inclusion and exclusion criteria. 

Point 6: Was there any standard for taking pictures by smartphone, for example resolution, light, distance,...?

Response 6: Thank you very much for the comment. Yes. We followed the standard protocols, such as the latest smartphone with the highest resolution and additional light sources, ensuring the visibility of the teeth. Furthermore, Photographs were taken at a standard distance of approximately 10 cm from the mouth to ensure a consistent field of view and detail, and the angle of the photographs was standardized to capture all relevant surfaces of the teeth.

Point 7: Are you sure proximal caries are detectable on photographs taken by smart phones?

Response 7: Thanks for the wonderful comment. We agree with the reviewer that capturing the proximal caries through the smartphone would be challenging. Nonetheless, we also faced the same challenge. However, the research team used one of the highest resolution cameras (smartphone) with appropriate angles to capture the caries and overcome this challenge. Nonetheless, as the reviewer commented, there might be a possibility of missing capture of the early stages of proximal caries. We included this as a limitation in the revised manuscript. 

Point 8: I could not find any description about severity of ECC in your manuscript. what do you mean of this word? How did you determine it by questionnaire?

Response 8: Thank you very much for the comment. The severity of ECC is included in the revised manuscript according to the reviewer’s comments.

Point 9: What was your sampling technique, convenient (line 86) or simple random (line 129)?

Response 9: Thanks for the comment and notifying us of the error. We used the convenience sampling method. We made the necessary changes in the revised manuscript.

Point 10: I could not find any explanation about type of ECC.

Response 10: Thanks for the comment. We included it in the methods section of the revised manuscript.

Point 11: I was interested to read about the dft index numbers in different groups in your study.

Response 11: Thank you for your valuable feedback and interest in the dmft index numbers in different groups. Primarily, after classifying the severity of the ECC according to the DMFT index, we combined all types of ECC and compared it with the absence of ECC to identify the determinants of ECC in this age group. This approach was planned to address the preventive public health strategies in all types of ECC. We indeed consider the reviewer’s suggestions for the planned future exploratory studies. 

Point 12: It is good. In table 3, "exckusive" word should be corrected.

Response 12: Thanks for the comment. We corrected it in the revised manuscript.

The authors thank the reviewer once again for the positive and constructive comments.

Reviewer 3 Report

Comments and Suggestions for Authors

Introduction

Line 29-31: please add the reference.

line 98: you would describe the questions in detail and provide if there is a new validation or a preexisting questionnaire.

table 1 is missing in the description. the tables must be understandable even without reading the full text.

line 142 the description should  follow the table as well as the description of Figure 1,2,3

figure 3 Please describe the meaning of type I, II or III of ECC potentially in the discussion section

I suggest rewriting the discussion section and adding more comments on your results, don't simply report literature findings

 references

please rewrite in the right order the references and be sure to cite all of them or delete no cited ones

Comments on the Quality of English Language

the language is correct, but could be improved in introduction section.

Author Response

Dear Reviewer,

Authors’ reply/modifications according to the reviewer 3 comments/suggestions

General:

The authors would like to thank the reviewer for the precious time spent reviewing the paper and his excellent suggestions for improving it. Efforts have been made to modify the paper as per the reviewer’s suggestions and recommendations. The authors will be happy to hear a positive reply. All the points included according to the reviewer’s comments can be seen in track changes.

Specific response to the reviewer’s suggestions:

Kindly find the attached response to each question one by one:

Point 1: Line 29-31: please add the reference.

Response 1: Thank you very much for the comment. The authors modified it according to the reviewer’s comments. Now, all references are incorporated through endnote software to ensure that all references are in the right order.

Point 2: line 98: you would describe the questions in detail and provide if there is a new validation or a preexisting questionnaire.

Response 2: Thank you very much for the comment. According to the reviewer’s comment, we provided more information about the data collection tool in the revised manuscript. The Cronbach Alpha value mentioned in the manuscript is for the new questionnaire. We modified the manuscript according to the reviewer’s comments for better understanding.

Point 3: table 1 is missing in the description.

Response 3: Thank you very much for the comment. Table 1 details can be found in the revised manuscript. If the reviewer needs further clarification, the research team is always happy to modify or revise the manuscript.

Point 4: line 142 the description should  follow the table as well as the description of Figure 1,2,3

Response 4: Thank you very much for the comment. The authors followed the description of tables/figures according to the reviewer’s comments in the revised manuscript.

Point 5: figure 3 Please describe the meaning of type I, II or III of ECC potentially in the discussion section

Response 5: Thank you so much for the comment. According to the reviewer's comments, we explained the types in the methods and results section.

For better clarification, the following statement is included in the methods section of the revised manuscript.

The decayed, missing, filled teeth (dmft) index was used to estimate the prevalence and severity of ECC in the study participants. The dmft index is specifically designed for primary teeth and is appropriate for assessing dental caries in children aged 36-71 months. We classified the severity of ECC based on the dmft index as follows: Mild ECC (type I) - dmft score of 1-3, Moderate ECC (type II)- dmft score of 4-6, and Severe ECC (type III) - dmft score of 7 or higher.

Point 6: I suggest rewriting the discussion section and adding more comments on your results, don't simply report literature findings

Response 6: Thank you very much for the comments. The discussion section is revised according to the reviewer’s suggestions.

Point 7: Please rewrite in the right order the references and be sure to cite all of them or delete no cited ones

Response 7: Thank you very much for the comments. The authors modified it according to the reviewer’s comments. Now, all references are incorporated through endnote software to ensure that all references are in the right order.

The authors thank the reviewer again for the positive and constructive comments.

Reviewer 4 Report

Comments and Suggestions for Authors

Dear Author,

After a thorough review, I believe that certain revisions could enhance the clarity and impact of your findings. Here are my suggestions:

  1. Clarification of Methodology: It would be beneficial for the reproducibility of the study if you could provide more detailed descriptions of the sampling methods and the criteria for the selection of participants. This would help in understanding the scope and limitations of the study more clearly.

  2. Expansion of the Discussion on Socioeconomic Factors: The role of socioeconomic factors is briefly mentioned; however, elaborating on how these factors specifically influence the prevalence and severity of ECC could provide deeper insights. For instance, discussing the economic barriers to accessing preventive dental care could add depth to the analysis.

  3. Inclusion of Comparative Data: Including comparative analyses with previous studies in similar settings could provide a more robust context for your findings. This comparison would help in understanding trends over time and geographical differences in ECC prevalence.

  4. Recommendations for Future Research: While the study concludes with preventive recommendations, suggesting areas for future research could be invaluable. For example, exploring the long-term outcomes of early dental interventions or the efficacy of various educational programs in different socioeconomic settings might be areas to consider.

  5. Enhancing Data Presentation: Some of the graphs and tables could be made more reader-friendly by using clearer labels and possibly reducing complexity. Simplifying these elements could make your important findings more accessible to a broader audience.

I hope you find these suggestions constructive. I look forward to seeing the revised version of your study, which no doubt will contribute significantly to the field and to improving public health strategies against ECC.

Regards

Author Response

Dear Reviewers,

Authors’ reply/modifications according to the reviewer 4 comments/suggestions

General:

The authors would like to thank the reviewer for the precious time spent reviewing the paper and his excellent suggestions for improving it. Efforts have been made to modify the paper as per the reviewer’s suggestions and recommendations. The authors will be happy to hear a positive reply. All the points included according to the reviewer’s comments can be seen in track changes.

Specific response to the reviewer’s suggestions:

Kindly find the attached response to each question one by one:

Point 1: After a thorough review, I believe that certain revisions could enhance the clarity and impact of your findings. Here are my suggestions:

Response 1: Thank you very much for the comment. The authors incorporated changes according to the reviewer’s comments. 

Point 2: Clarification of Methodology: It would be beneficial for the reproducibility of the study if you could provide more detailed descriptions of the sampling methods and the criteria for the selection of participants. This would help in understanding the scope and limitations of the study more clearly.

Response 2: Thank you very much for the comments. We agree with the reviewer; the authors have made the necessary changes to the revised manuscript. The present study used a convenience sampling method from a single center. We also added the details in the limitations according to the reviewer’s comments. More details are included in the revised manuscript. Furthermore, more details about the data collection tool are included in the revised manuscript in accordance with the reviewer’s comments.

Point 3: Expansion of the Discussion on Socioeconomic Factors: The role of socioeconomic factors is briefly mentioned; however, elaborating on how these factors specifically influence the prevalence and severity of ECC could provide deeper insights.

Response 3: Thank you very much for the insightful comments. The authors included discussion aspects that emphasize socioeconomic factors and their influence on the prevalence and severity of ECC with additional references.

Point 4: Inclusion of Comparative Data: Including comparative analyses with previous studies in similar settings could provide a more robust context for your findings. This comparison would help in understanding trends over time and geographical differences in ECC prevalence.

Response 4: Thanks for the comments. The authors included more studies in the revised manuscript that help to compare the trends and variation across the regions.

Point 5: Recommendations for Future Research: While the study concludes with preventive recommendations, suggesting areas for future research could be invaluable. For example, exploring the long-term outcomes of early dental interventions or the efficacy of various educational programs in different socioeconomic settings might be areas to consider

Response 5: Thanks for the valuable comments. According to the reviewer’s comments, we included recommendations for early dental intervention and evaluation of these programs for effective preventive strategies.

Point 6: Enhancing Data Presentation: Some of the graphs and tables could be made more reader-friendly by using clearer labels and possibly reducing complexity. Simplifying these elements could make your important findings more accessible to a broader audience.

Response 6: Thanks for the comment. We modified it according to the reviewer’s comment.

Point 7: I hope you find these suggestions constructive. I look forward to seeing the revised version of your study, which no doubt will contribute significantly to the field and to improving public health strategies against ECC.

Response 7: Thanks for the positive and constructive comments. The authors also firmly trust that the revised version will significantly help improve public health strategies against ECC.

The authors thank the reviewer again for the positive and constructive comments.

Round 2

Reviewer 2 Report

Comments and Suggestions for Authors

Thanks for the changes you made by revision.

Reviewer 3 Report

Comments and Suggestions for Authors

Dear author,

thank you for your efforts, I think the article is now ready to publish 

Reviewer 4 Report

Comments and Suggestions for Authors

I am writing to express my satisfaction with the revisions you have made to [Early Childhood Caries: Prevalence, Associated Factors, and Severity: A Hospital-Based Study from Riyadh, Saudi Arabia]. After reviewing the updated version, I am pleased to confirm that the changes have effectively addressed the concerns and suggestions previously discussed.

Your attention to detail and commitment to improving the quality of the work is highly commendable. The revisions have enhanced the clarity and overall coherence, aligning perfectly with the intended objectives.

Thank you for your diligence and dedication throughout this process. I appreciate your efforts and look forward to any future collaborations.